# Rates of morphological evolution in Captorhinidae: an adaptive radiation of Permian herbivores

Neil Brocklehurst

Museum für Naturkunde, Leibniz-Institut für Evolutions- und Biodiversitätsforschung, Berlin, Germany

## ABSTRACT

The evolution of herbivory in early tetrapods was crucial in the establishment of terrestrial ecosystems, although it is so far unclear what effect this innovation had on the macro-evolutionary patterns observed within this clade. The clades that entered this under-filled region of ecospace might be expected to have experienced an "adaptive radiation": an increase in rates of morphological evolution and speciation driven by the evolution of a key innovation. However such inferences are often circumstantial, being based on the coincidence of a rate shift with the origin of an evolutionary novelty. The conclusion of an adaptive radiation may be made more robust by examining the pattern of the evolutionary shift; if the evolutionary innovation coincides not only with a shift in rates of morphological evolution, but specifically in the morphological characteristics relevant to the ecological shift of interest, then one may more plausibly infer a causal relationship between the two.

Here I examine the impact of diet evolution on rates of morphological change in one of the earliest tetrapod clades to evolve high-fibre herbivory: Captorhinidae. Using a method of calculating heterogeneity in rates of discrete character change across a phylogeny, it is shown that a significant increase in rates of evolution coincides with the transition to herbivory in captorhinids. The herbivorous captorhinids also exhibit greater morphological disparity than their faunivorous relatives, indicating more rapid exploration of new regions of morphospace. As well as an increase in rates of evolution, there is a shift in the regions of the skeleton undergoing the most change; the character changes in the herbivorous lineages are concentrated in the mandible and dentition. The fact that the increase in rates of evolution coincides with increased change in characters relating to food acquisition provides stronger evidence for a causal relationship between the herbivorous diet and the radiation event.

Corresponding author
Neil Brocklehurst,
neil.brocklehurst@mfn-berlin.de

## INTRODUCTION

The evolution of high fibre herbivory represents a major step in the establishment of terrestrial ecosystems. Prior to the appearance in the Pennsylvanian of tetrapods capable of feeding directly on plant matter, the vast majority of primary consumers in the terrestrial realm are thought to have been detritivorous invertebrates (*Shear & Sheldon, 2001*). By the end of the Cisuralian, five tetrapod lineages had independently evolved a herbivorous diet and terrestrial ecosystems were adopting a more modern set of tropic interactions, with

a great abundance of large terrestrial vertebrates supporting a relatively small number of macro-carnivores (*Olson, 1966*; *Sues & Reisz, 1998*).

Although arthropod herbivores were present in terrestrial ecosystems prior to the evolution of herbivory in tetrapods, evidence both of body fossils and feeding damage to plants is rare (*Labandeira, 1998*; *Labandeira, 2007*; *Shear, 2000*). In a review of the Carboniferous flora of Mazon Creek, *Scott & Taylor (1983)* found that only 4% showed any sign of having been bitten or chewed by arthropods. Terrestrial vertebrate herbivores were entering a somewhat under-filled region of ecospace, facing little competition. These early herbivores therefore provide an ideal opportunity to examine the changes in rate and mode of evolution and diversification resulting from evolutionary innovations. Simpson's adaptive radiation model (*Simpson, 1953*) refers to the rapid emergence from a common ancestor of many species, coinciding with both ecological and phenotypic divergence of the descendants. Simpson posited that a "key" evolutionary novelty gives a lineage a selective advantage or allows it to enter a new ecological niche and thus leads to an increase in morphological diversification or speciation rates. Such a model is often invoked when analyses of diversification rate heterogeneity identify shifts that coincide with an innovation of interest (e.g., *Benson & Choiniere, 2013*; *Cook & Lessa, 1998*; *Forest et al., 2007*; *Kazancıoğlu et al., 2009*; *Kozak et al., 2005*; *McLeish, Chapman & Schwarz, 2007*; *Ruber, Van Tassell & Zardoya, 2003*; *Vences et al., 2002*).

Since this seminal work, a substantial number of studies have refined our understanding of the evolutionary processes behind adaptive radiations. It has now been acknowledged that adaptive radiations do not necessarily require rapid speciation (*Neige, Dera & Dommergues, 2013*; *Givnish, 2015*; *Simões et al., 2016*); increases in rates of morphological diversification may take place in the absence of increases in species richness and rates of speciation. The concept of the "key innovation", critical to Simpson's model, has come under scrutiny. Often these innovations are not a single trait, but instead a stepwise acquisition of traits or co-option of existing traits for a novel purpose (exaption) that allows the entry into a new region of ecospace (*Donoghue, 2005*; *Simões et al., 2016*). Thus, the radiation may not coincide with the acquisition of a relevant trait (*Lieberman, 2012*). Models of radiations dependant on diversity and abundance have been examined: colonists of new or depopulated adaptive landscapes where there is little competition would maximise selection for divergence (*Givnish, 2010*; *Givnish, 2015*). By extension, as the radiation fills the new region of ecospace, the rate of diversification should slow: an early burst model (*Blomberg, Garlan & Ives, 2003*). However, examination of empirical data has produced conflicting results regarding the ubiquity of an early burst model: *Harmon et al. (2010)* suggested that early bursts were rare when examining size and shape data, but when examining discrete character data *Hughes, Gerber & Wills (2013)* found early peaks in disparity followed by diversification slow-downs or decreases were the norm.

The inference of a causal relationship between innovation and shift in diversification rate is in most cases circumstantial, based solely on the coincidence of the shift and the evolutionary novelty. In order to more reliably infer a causal relationship, one must also examine the precise nature of the shift. For example, *Brocklehurst et al. (2015)* showed that, although early amniotes do exhibit lineage diversification rate increases coinciding with

various "key" innovations, the shifts did not represent increases in speciation rate but instead coincided with periods of increased extinction rate. Thus, it was inferred that these innovations did not cause Simpsonian adaptive radiations, but instead buffered against high levels of extinction.

In the same way, when attempting to infer a causal relationship between a key innovation and a shift in rates of morphological evolution, it is not enough to point to a rate shift along the branch where the innovation appeared, but one must also examine the morphological changes occurring subsequent to the shift: is the clade of interest showing a higher rate of change in features relevant to the exploitation of the new ecological niche allowed by the key innovation? If not, there is unlikely to be a causal link between the two.

This logic is here applied to an examination of rates of morphological evolution in the earliest herbivores, using the family Captorhinidae as a case study. Captorhinids were a diverse clade of sauropsids (reptile-line amniotes) that appeared during the Late Pennsylvanian (*Müller & Reisz, 2005a*; *Müller & Reisz, 2005b*) and survived until the end of the Permian. Herbivorous members of this clade appear in the Kungurian, characterised by the multiple rows of teeth and a propalineal motion of the lower jaw in order to grind and shred plant matter (*Dodick & Modesto, 1995*; *Modesto et al., 2007*). In this paper, I examine the rates of morphological evolution in this family using a method incorporating a time calibrated phylogeny and a matrix of discrete characters (*Lloyd, Wang & Brusatte, 2012*). Emphasis is placed not only on examining whether rate increases coincide with shifts in diet, but also on examining whether a shift in diet coincides with increased frequency of character-state transformation in regions related to feeding, such as the dentition. In this way, a more robust inference may be made concerning the possibility of an adaptive radiation coinciding with the origin of herbivory in this family.

# MATERIALS AND METHODS

## Phylogeny and time calibration

The phylogeny used was that which was presented in *Leibrecht et al. (in press)*, currently the most comprehensive cladistic analysis of captorhinids. The phylogeny was time calibrated in the R 3.1.2 (*R Core Team, 2014*) using the method proposed by *Lloyd et al. (2016)*, itself an expansion of a method put forward by *Hedman (2010)*. The method of *Hedman (2010)* was intended to infer confidence intervals on the age of a specific node in the tree. It is a Bayesian approach using the ages of successive straigraphically consistent outgroup taxa relative to the age of the node of interest to make inferences about the quality of sampling; large gaps between the age of the node of interest and that of the outgroups implies a poorly sampled fossil record, and therefore the age of the node of interest may be inferred to be older. *Lloyd et al. (2016)* designed a procedure whereby this approach could date an entire tree rather than just a specific node. In applying this method, successive outgroups are required to the total clade. The outgroups to Captorhinidae employed were: *Paleothyris* and *Hylonomus* (found to be the outgroups to Captorhinidae in the Bayesian analyses of *Müller & Reisz (2005a)* and *Müller & Reisz (2005b)*), *Archaeothyris* (the earliest known synapsid (*Reisz, 1972*)), and *Westlothiana* (a reptiliamorph outside the amniote crown according to

Ruta & Coates, 2007). A maximum age constraint was set as 334.7 million years ago, the oldest reliable estimate using molecular dating for the origin of Amniota published within the last five years at the moment of data collection (Parfrey et al., 2011).

Uncertainty surrounding the ages of taxa was accounted for using the method of Pol & Norell (2006). For each taxon (including the outgroups), 100 first appearances and last appearances were drawn at random from a uniform probability distribution covering the full possible range of ages for that taxon. One hundred time-calibrated trees were produced from the 100 sets of ages.

Since the analysis of Leibrecht et al. (in press) produced two most parsimonious trees (MPTs), half the time calibrated trees were based on the first, and half on the second. All analyses described below were carried out on all 100 of these trees. All 100 of these trees are available in Data S1, and the age ranges allowed for each taxon in Data S2.

## Reconstruction of dietary evolution

A dietary character with three states, carnivore, herbivore and omnivore, was scored for all taxa present in the phylogeny. Ancestral character states were deduced using conditional (joint) likelihood, employing the *ace* function in the ape package (Paradis, Claude & Strimmer, 2004) in R. This function allow three models of discrete character change: an equal rates model (transitions between all states in all directions are equally probable), a symmetrical model (transitions between two character states occur with equal probability in either direction, but different pairs of character states have different probabilities of transition) and an all-rates-different model (each transition has a different probability). In order to deduce which model was best for the data available, these models were fit to the captorhinid phylogeny using likelihood methods, employing the *fitDiscrete* function in the R package geiger (Harmon et al., 2008). The Akaike weights were used to deduce the best fitting model.

Since this likelihood ancestral state reconstruction produced uncertain results surrounding the character state at three nodes (see below), an alternative Bayesian approach for optimizing the dietary character at these nodes was used as an independent test in the program BayesTraits V2 (Barker, Meade & Pagel, 2007). A reversible jump Markov chain Monte Carlo (RJ MCMC) approach was used to reconstruct the ancestral states of the discrete character, allowing different combinations of character states to have different transition rates. The analysis was run for 10,000,000 iterations with 100,000 discarded as burn-in, sampling every 1,000 generations. These results are presented in the Data S1, and the analyses described below are based on the likelihood results.

## Analysis of rate variation

Analysis of rate variation was carried out using the method of Lloyd, Wang & Brusatte (2012), later refined by Brusatte et al. (2014) and Close et al. (2015). Discrete morphological character scores may be taken from the matrices used in cladistic analyses, and ancestral states are deduced using likelihood. This allows the number of character changes along each branch to be counted, and rates of character change are calculated by dividing the number of changes along a branch by the branch length. The absolute value calculated

for the rate of each branch, however, can be misleading due to the presence of missing data (*Lloyd, Wang & Brusatte, 2012*). As such it is more useful to identify branches and clades where the rates of character change are significantly higher or lower than others, rather than comparing the raw numbers. This is assessed by comparing two models using a likelihood ratio test, one where the rates of change are uniform across the whole tree and one where the branch of interest has a different rate to the rest of the tree. A similar method is used to compare rates of evolution through time and identify bins where rates of evolution are significantly high or low.

The character data used was from the matrix of *Leibrecht et al. (in press)*. The time bins used to examine rate variation through time were substages, dividing the international stages into two bins: early and late. The analysis was carried out in R using functions from the package Claddis (*Lloyd, 2016*) on all 100 of the time calibrated trees. The data matrix is presented in Data S3.

Due to the uncertainty surrounding the optimisation of the dietary character, a stochastic mapping approach was used to examine rate heterogeneity in the different dietary classes. For each of the 100 time calibrated trees, the dietary character containing three states (carnivore, omnivore and herbivore) was mapped onto the tree using likelihood. Using the character state probabilities identified at each node, 100 possible evolutionary histories of diet in that tree were generated for each of the 100 phylogenies following the procedure outlined by *Bollback (2006)*, giving a total of 10,000 stochastic maps. The mean rate of herbivorous branches, carnivorous branches and omnivorous branches were calculated in each stochastic map, along with the mean rate of a randomly selected set of branches with a sample size equal to the number of herbivorous branches in that map.

Taxonomic jack-knifing was used as a sensitivity analysis to examine the robusticity of the results. In each iteration, three randomly selected taxa (the maximum number that could be deleted leaving at least one member of each dietary regime) were dropped prior to running the analyses of rate variation described above.

## Disparity

The character matrix of *Leibrecht et al. (in press)* was also used to examine morphological diversity (disparity). Morphological distances between taxa were calculated using the Maximum Observable Rescaled Distance (MORD) distance measure of *Lloyd (2016)*, which was shown to perform better in datasets with large amounts of missing data. Following the suggestion of *Brusatte et al. (2011)*, the internal nodes of the phylogeny were treated as data points, with their character scores inferred using ancestral state reconstruction, in order to account for the incomplete sampling of the fossil record; these data points represent ancestral taxa that may have possessed character combinations not observed in sampled taxa.

Having generated a distance matrix, once again the stochastic mapping approach was used to compare disparity in different dietary classes. For each of the 10,000 evolutionary histories generated, each taxon (both tip and node) was assigned a dietary class, and the mean MORD distance for each of the three dietary classes was calculated.

Disparity through time was investigated by subjecting the MORD distance matrices to a principal coordinate analysis. Disparity in each time bin was calculated as the sum of variances of the PC scores of each taxon in that bin. An attempt was also made to incorporate both ghost lineages and internal branches into the analysis using a novel method illustrated in Fig. 1. Taxon A is present in time bin 3, and its ancestral node is inferred to be in time bin 1. Therefore there must be a ghost lineage present in time bin 2 (Fig. 1A), which would be ignored in the disparity analysis under the method of *Brusatte et al. (2011)*, wherein only node and tip morphologies were included. The morphology inferred in time bin 2 will depend on which model of evolution is preferred; under a gradualistic model of evolution, assuming no change in rate along the branch (Fig. 1B), the principal coordinate score in time bin 2 may be inferred by calculating the rate of change in the principal coordinate along that branch, and the amount of time between the ancestral node and the midpoint of time bin 2. Alternatively one may assume a punctuated model of evolution, where the morphological change occurs rapidly at the time of speciation in time bin 1, and the lineage experiences morphological stasis for the remaining time; thus the PC scores inferred in time bin 2 will be identical to that of the tip in time bin 3 (Fig. 1C). Both methods are used here to compare the results. Again, the stochastic mapping approach was used to assign a diet to each branch, allowing the comparison of patterns of disparity through time in each of the dietary regimes. The function used to infer trait disparity through time while including ghost lineages is presented in Data S5.

## Lineage density

*Sidlauskas (2008)* highlighted that rate heterogeneity is not the only means by which different clades may have different disparities. The "efficiency" with which the taxa explore morphospace will also have an influence. A clade which continually returns to the same region of morphospace will exhibit a lower disparity than a clade that explores novel regions of morphospace, even if the rates of morphological change do not vary (*Sidlauskas, 2008*). In fact, it has been observed that increased disparity can occur even alongside decreases in evolutionary rates; when examining the evolution of mammals across the Cretaceous/Paleogene boundary, *Slater (2013)* demonstrated that, although rates of body size evolution were lower after the boundary, an increase in variance was possible due to the release of evolutionary constraint.

*Sidlauskas (2008)* introduced a method to examine this concept: Lineage Density. This is a measure of the amount of evolutionary change within a clade relative to the area of morphospace explored. It is calculated by dividing the sum of the morphometric branch lengths within a clade by the volume of the 95% confidence hyperellipsoid or convex hull of the clade's morphospace. A lower lineage density indicates that the taxa are exploring novel regions of morphospace rather than continuously returning to the same morphologies.

The principal coordinate analysis described above was used as the basis for calculating lineage density. For each of the 10,000 dietary histories generated by stochastic mapping, the branch lengths within each dietary regime were calculated as the Euclidean distance between each data point i.e., the morphological distance travelled by that branch. The 95% confidence hyperellipsoid volume of each dietary regime was calculated using the functions

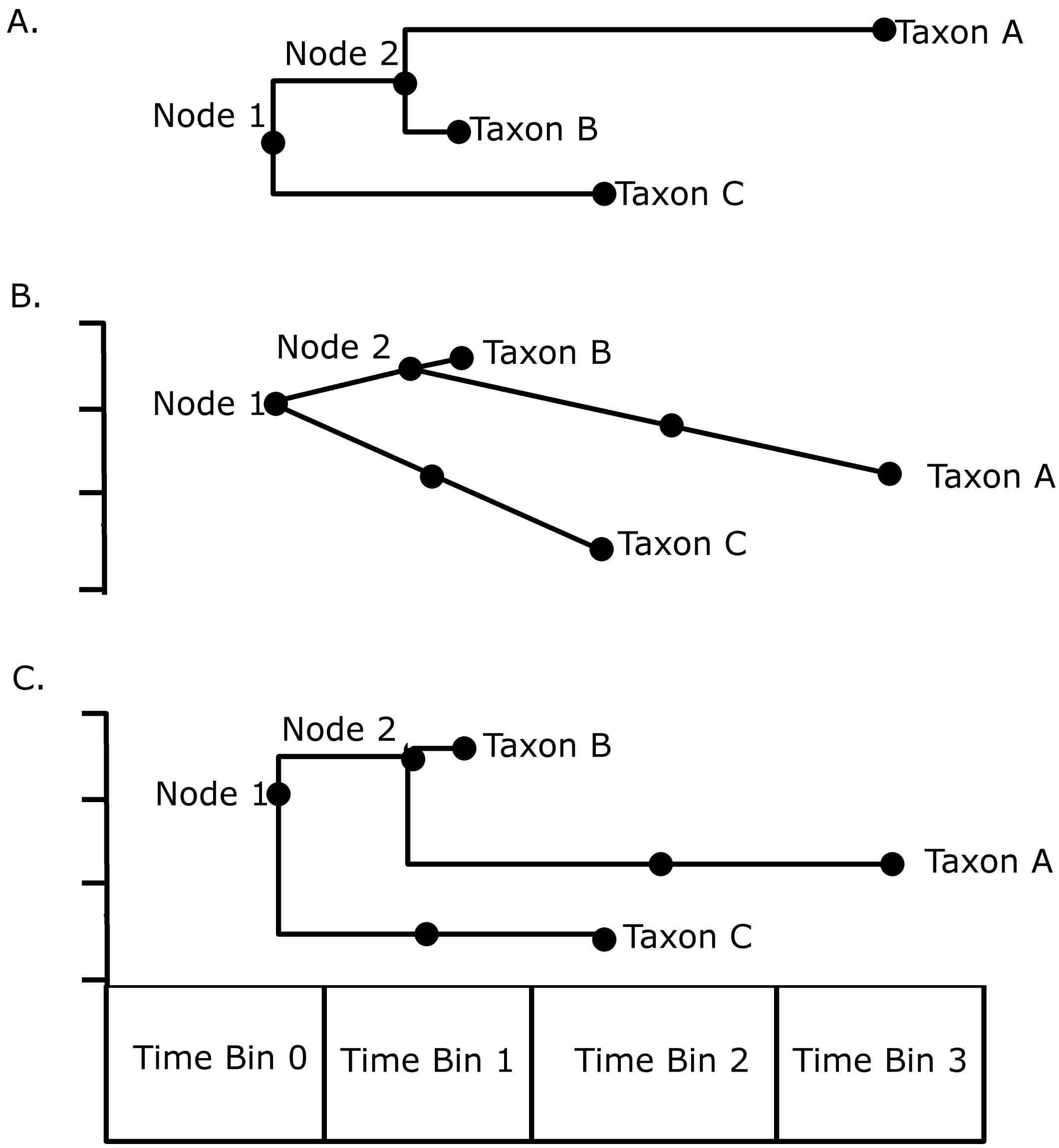

**Figure 1 An illustration of the methods used to calculate disparity in this study.** (A) A hypothetical phylogeny illustrating as solid dots the data points that would be included under the method of *Brusatte et al. (2011)*: the tip taxa A, B and C, and the Nodes 1 and 2; (B) The phylogeny plotted against a hypothetical trait, illustrating how the morphology of the lineage leading to Taxon C in time bin 1 and the morphology of the lineage leading to taxon A in time bin 2 may be inferred assuming a gradual model of evolution with no rate variation along a branch; (C) An illustration of how the same morphologies are inferred assuming a punctuated model of evolution, where the morphological change occurs at the speciation event.

in the R package Cluster *Maechler et al. (2008)*. The lineage density of each dietary regime was calculated using the equation provided by *Sidlauskas (2008)*.

### Character change histories

The character list of *Leibrecht et al. (in press)* was divided into five categories based on the region to which the characters referred to: Skull, Palate, Mandible, Dentition and Postcranium. The functions in the package Claddis automatically calculates the most likely combination of character changes for each of the 100 time calibrated phylogenies alongside the analysis of rate variation. These character change histories were used to assess which region of the skeleton underwent the greatest change within each dietary regime. Using the 10,000 stochastic maps of the dietary character, the number of characters from each region changing within each dietary regime was counted. These counts for each region were divided by the total number of character changes occurring across the entire tree in that region to account for the fact that the characters were not evenly distributed. The list of characters and the region to which they were assigned is presented in Data S4.

## RESULTS

### Dietary evolution

Fitting of discrete models of character evolution to the dietary character indicates an equal rates model best fits the captorhinid phylogeny (Fig. 2). It should be noted that this support is not overwhelming; although the ER model is found to fit best in all 100 trees, in none does it receive an Akaike weights score above 0.8. Using this model in ancestral state reconstructions (Fig. 3) indicates that a single transition to a herbivorous diet is most probable. *Labidosaurus*, judged to be an omnivore by *Modesto et al. (2007)* on the basis of the dental morphology, is found to most likely have evolved from a herbivorous ancestor, rather than *Captorhinikos chozaensis* and the Moradisaurinae representing convergent transitions to herbivory from an omnivorous ancestor. There is, however, considerable uncertainty; the probability of a herbivorous ancestor is not much more than 50%. There is further uncertainty surrounding the ancestral diet of the clade containing the three species of *Captorhinus*, *Captorhinikos chozaensis*, *Labidosaurus* and the Moradisaurinae; while an omnivorous ancestor receives the highest likelihood, the probability is not much better than that of a carnivorous ancestor. This has implications for the transition to herbivory: the transition from carnivory to herbivory may have passed through an omnivorous phase, which was retained by the genus *Captorhinus* (*Dodick & Modesto, 1995*; *Kissel, Dilkes & Reisz, 2002*), or the genus *Captorhinus* may represent a transition to omnivory from carnivory independent of the transition to herbivory.

For the three uncertain nodes, the ancestral state reconstruction using the RJ MCMC method produced similar results. In fact this approach suggested less uncertainty surrounding the ancestral states (Fig. S1). Thus the most probable evolutionary history inferred is a single transition to herbivory via omnivory, with *Labidosaurus* representing a reversal to an omnivorous diet from a herbivorous ancestor.

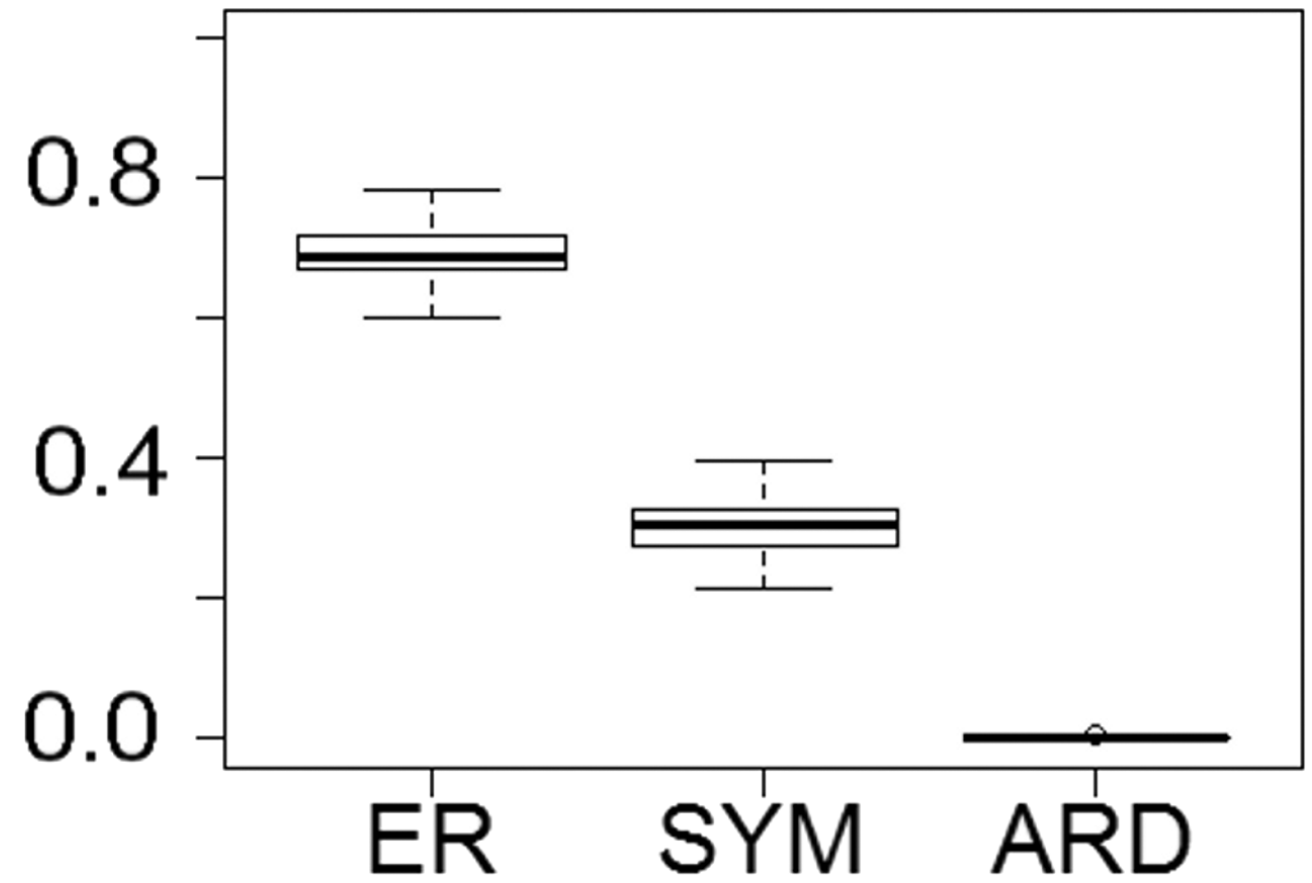

**Figure 2 The fit of models of diet evolution to the phylogeny of Captorhinidae.** Boxplots illustrating the distribution of 100 Akaike weights values calculated for each of the models of the evolution of diet as a discrete character, fit to the 100 time calibrated phylogenies of captorhinids. ER, Equal Rates; SYM, Symmetrical; ARD, All Rates Different.

## Analyses of rate heterogeneity

In the overwhelming majority of the 100 time calibrated trees, a significant rate increase is identified along the branch leading to the Moradisaurinae (Fig. 4), the clade containing exclusively herbivorous taxa. This result is robust with taxonomic jack-knifing: in 94% of the jack-knife iterations a significant rate increase is identified along this branch. The position of other significant increases in rate depends on the tree topology and the uncertainty in dating the taxa, but in more than half of the trees the branch leading to the clade containing the Moradisaurinae, *Labidosaurus* and *Captorhinikos chozensis* (the clade inferred to have a herbivorous ancestor) are found to exhibit a rate increase, as is the lineage leading to the clade containing *Labidosaurus* and Moradisaurinae in more than two thirds of the trees. Significant rate decreases are observed in the lineages leading to *Saurorictus* and to *Labidosaurus* in the majority of the 100 trees.

While the analyses did identify rate heterogeneity when comparing branches of the phylogeny, when comparing rates of evolution in different time bins, very little was identified. In all of the 100 time calibrate trees, a constant rate through time was found to have a higher likelihood than a different rate in each time bin.

## Rates and disparity in different dietary regimes

Of the 10,000 stochastic maps of dietary evolution in captorhinids, the herbivores have a higher mean rate of discrete character change than the omnivores in 9,845 maps, and a higher rate than the carnivores in 9,986 (Fig. 5A). When the mean rates of herbivores are compared to an equal number of branches drawn at random, the herbivores have a higher mean rate in 9,484 of the stochastic map (Fig. 5B). This result is robust with taxonomic jack-knifing: in 98% of jack-knife iterations the mean rates of character change in herbivores are higher than the carnivores, and are higher than the omnivores in 97% of the iterations (Fig. S2). In all 10,000 stochastic maps, the mean morphological distance

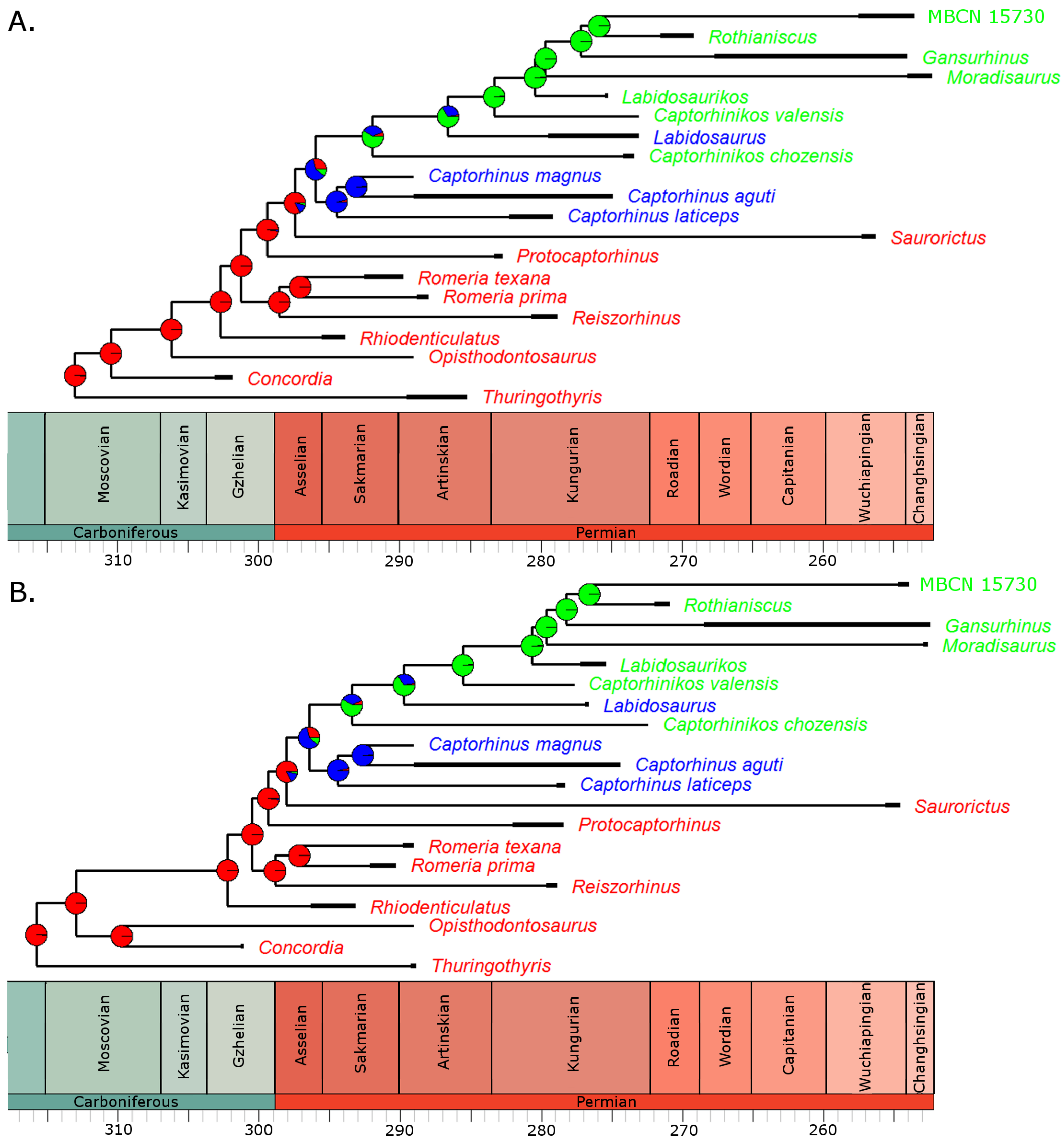

**Figure 3 The phylogeny of Captorhinidae, illustrating the evolution of diet.** Two of the 100 time calibrated phylogenies used in the analysis. The thick branches represent the observed ranges of each taxon. The colours of the tip labels represent the diet inferred for that taxon: Red, Carnivore; Blue, Omnivore; Green, Herbivore. The pie charts at each node represent the probability of each dietary regime inferred for that node, deduced by maximum likelihood ancestral state reconstruction. (A) MPT 1: *Opisthodontosaurus* is the sister to the clade containing *Rhiodenticulatus* and all captorhinids more derived. (B) MPT 2: *Opisthodontosaurus* is the sister to *Concordia*.

between the herbivorous taxa is greater than that of the omnivores and the carnivores, indicating a greater disparity (Fig. 6).

## Disparity through time

When evolutionary change is assumed to be gradual (Fig. 7A), the carnivorous captorhinids show a gradual increase in morphological disparity up to a peak in the early Artinskian. Through the late Artinskian and Kungurian their disparity decreases, culminating in a

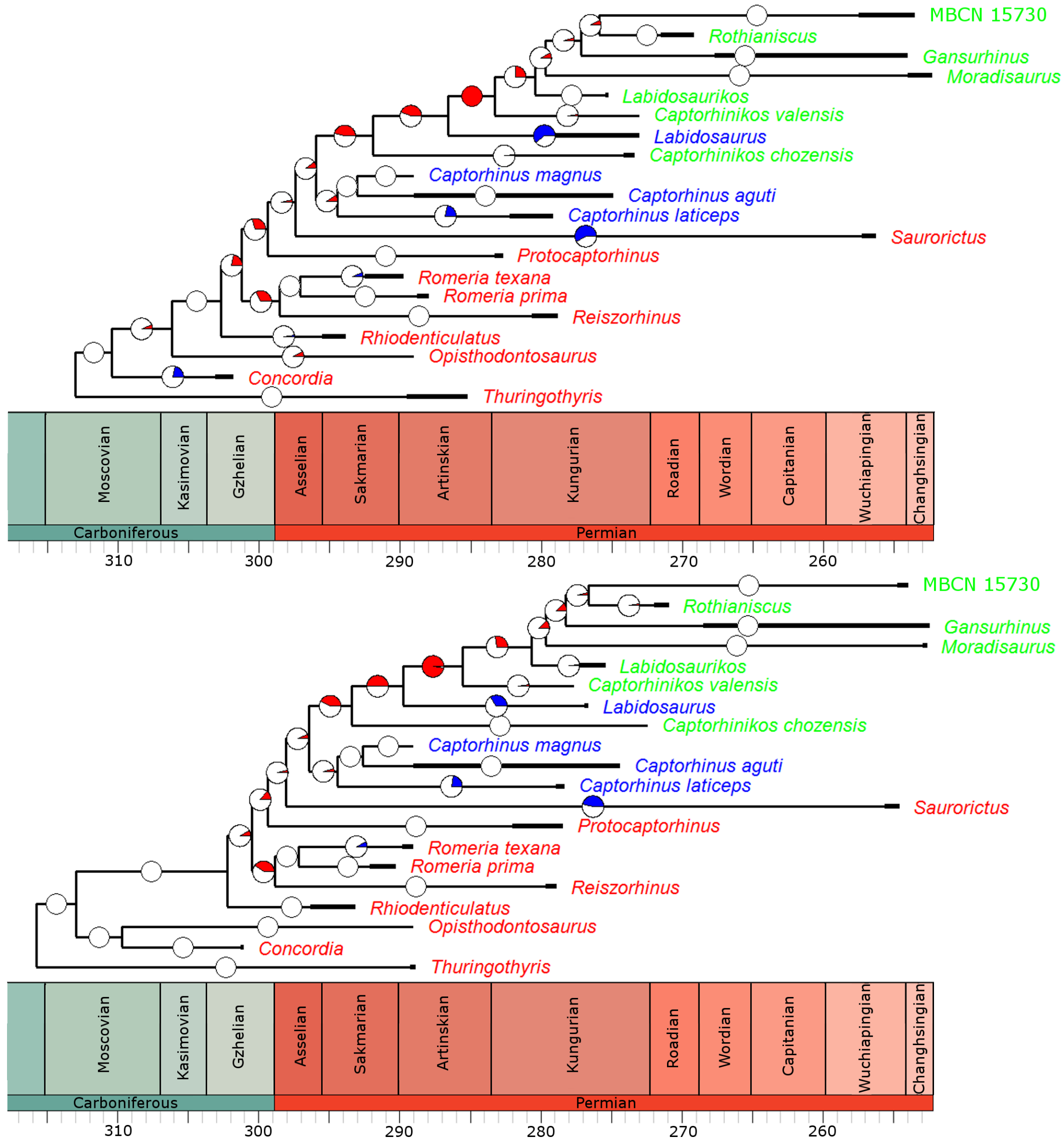

**Figure 4 The phylogeny of Captorhinidae, illustrating the location of significant changes in rates of evolution.** Two of the 100 time calibrated phylogenies used in the analysis. The thick branches represent the observed ranges of each taxon. The colours of the tip labels represent the diet inferred for that taxon: Red, Carnivore; Blue, Omnivore; Green, Herbivore. The pie charts on each branch represent the proportion of the 100 time calibrated phylogenies which show significantly high or low rates of evolution along that branch: Red, significantly high rates; Blue, significantly low rates; White, no significant rate variation. (A) MPT 1: *Opisthodontosaurus* is the sister to the clade containing *Rhiodenticulatus* and all captorhinids more derived. (B) MPT 2: *Opisthodontosaurus* is the sister to *Concordia*.

fall to zero across the Kungurian/Roadian boundary, after which only one carnivorous captorhinid is included in the phylogeny (*Sauroricitus*). The omnivorous captorhinids show a similarly gradual increase in disparity between the Asselian and Kungurian. Again, their disparity falls to zero across the Kungurian/Roadian boundary. The initial establishment of the disparity of the herbivorous lineages is more rapid than that of the carnivores, having exceeded the disparity of the carnivorous captorhinids by the early Kungurian. A

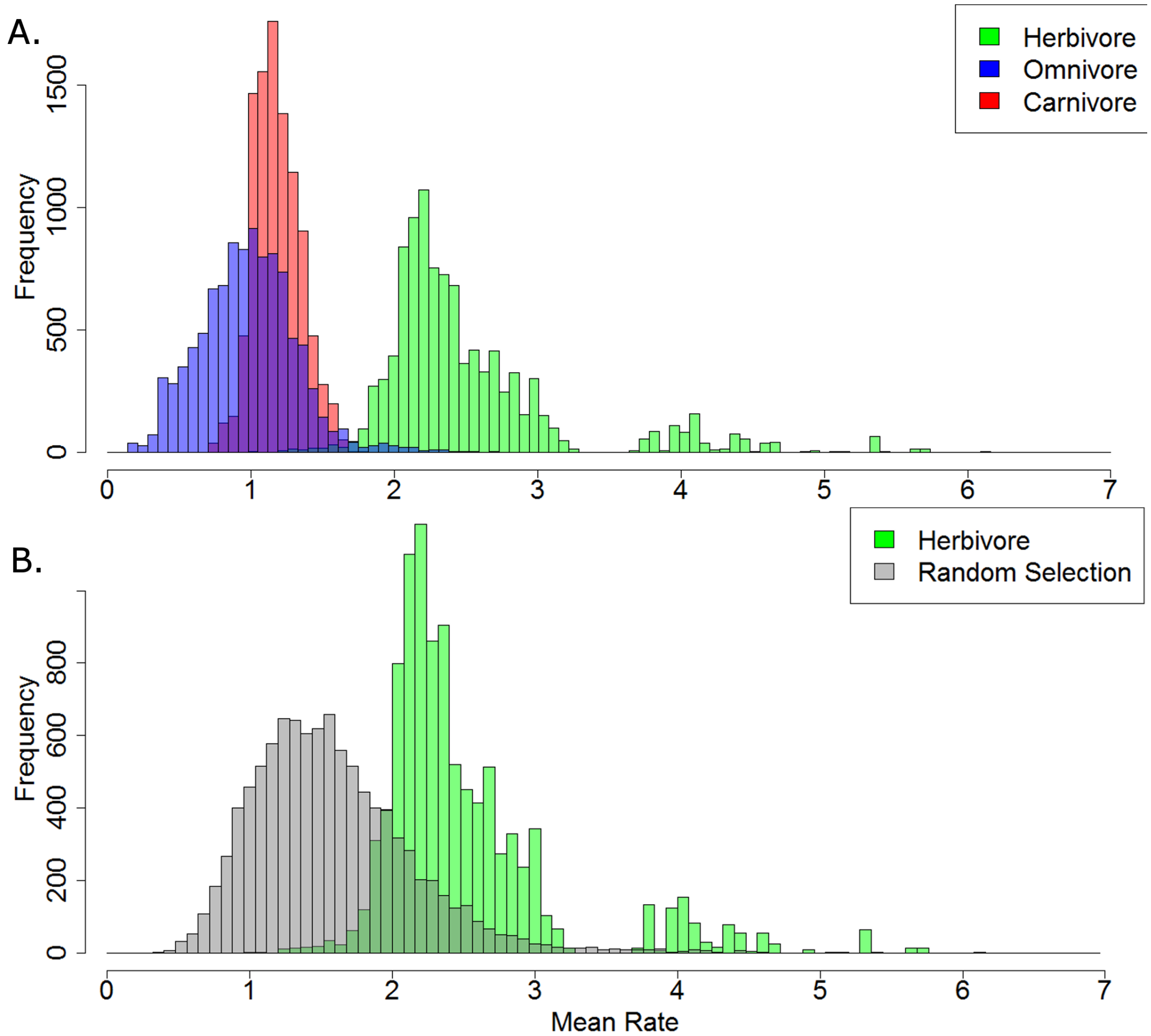

**Figure 5 A comparison of the mean rates of evolution within each dietary regime.** (A) Histogram illustrating the mean rate of discrete character evolution calculated for each dietary regime in each of the 10,000 stochastic maps of dietary evolution; (B) Histogram illustrating the mean rate of discrete character evolution calculated for the herbivorous lineages compared to a random selection of branches with an equal sample size in each of the 10,000 stochastic maps of dietary evolution.

disparity peak is reached in the late Kungurian, higher than the peaks observed either in the carnivorous or omnivorous curves. Herbivore disparity falls across the Kungurian/Roadian boundary, but recovers by the Wuchiapingian.

If morphological change is assumed to be punctuated, with the morphological change occurring at the speciation events (Fig. 7B), then in all three dietary classes peak morphological disparity is reached soon after that regime's appearance and disparity remains fairly constant in the bins following. As observed when using the gradualistic model, however, peak disparity of the herbivores is higher than either the omnivores or the carnivores. Interestingly, the disparity of herbivores already exceeds that of the other two dietary regimes by the Artinskian when using the punctuated model. Moreover, the decrease in disparity observed in the herbivorous lineages across the Kungurian/Roadian boundary is of a much lesser extent and disparity has recovered by the Wordian.

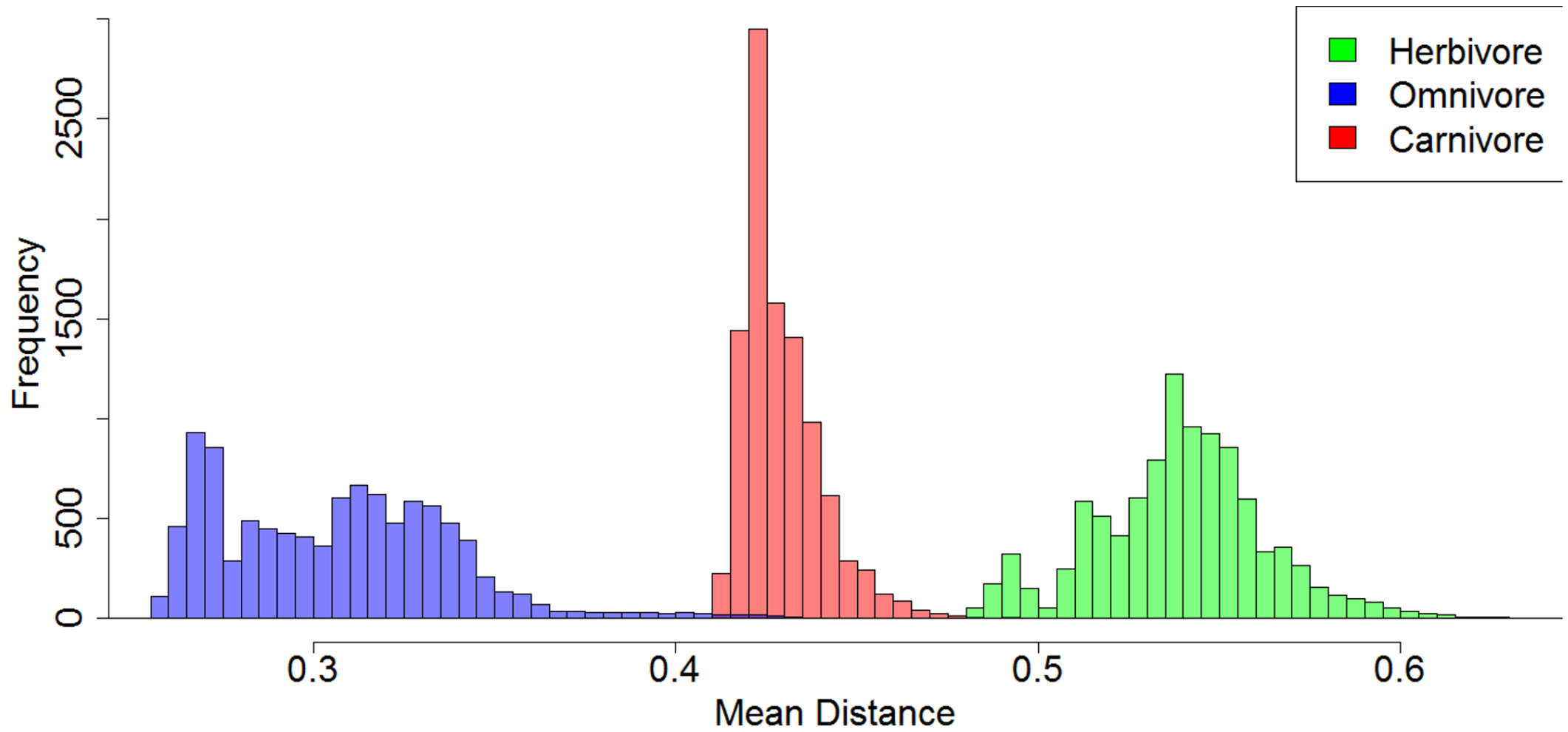

**Figure 6** **A comparison of the morphological distances between taxa within each dietary regime.** Histogram illustrating the mean MORD distance between each taxon in each each dietary regime in each of the 10,000 stochastic maps of dietary evolution.

## Lineage density

While there is considerable variation in the absolute lineage densities calculated within each dietary regime (Fig. 8), the overwhelming majority of the 10,000 evolutionary histories examined show the same relative pattern. In 9,926 of these, the lowest lineage density is found in the herbivores, while the highest lineage density is observed in the omnivores. In only 58 of the 10,000 stochastic maps do the carnivores have a lower lineage density than the herbivores.

## Character change histories

The majority of character changes in the carnivorous lineages occurred in the skull and postcranium (Fig. 9A). In most of the 10,000 stochastic maps the feeding apparatus (teeth and mandible) remain more conservative, with a lower proportion of character changes occurring in these regions. This changes with the transition to herbivory: the majority of the characters changing in herbivorous captorhinids are dental characters and, in many of the stochastic maps (but not all), mandibular characters (Fig. 9C). The postcranium and skull, the most plastic regions in the carnivorous captorhinids, show lower proportions of character change in this new dietary regime. There is little difference in the proportions of characters changing in each region in the omnivorous captorhinids (Fig. 8B).

# DISCUSSION

The link between a supposed "key innovation" and an adaptive radiation must always, to a certain extent, be circumstantial; one may identify the branch in a phylogeny along which the evolutionary novelty likely appeared, and one may identify the location of shifts in rates of evolution and diversification, but conclusively proving a causal relationship between the two is extremely difficult. Nevertheless, the evidence supporting an adaptive radiation of captorhinids coinciding with the origin of herbivory in this clade is compelling. It is only

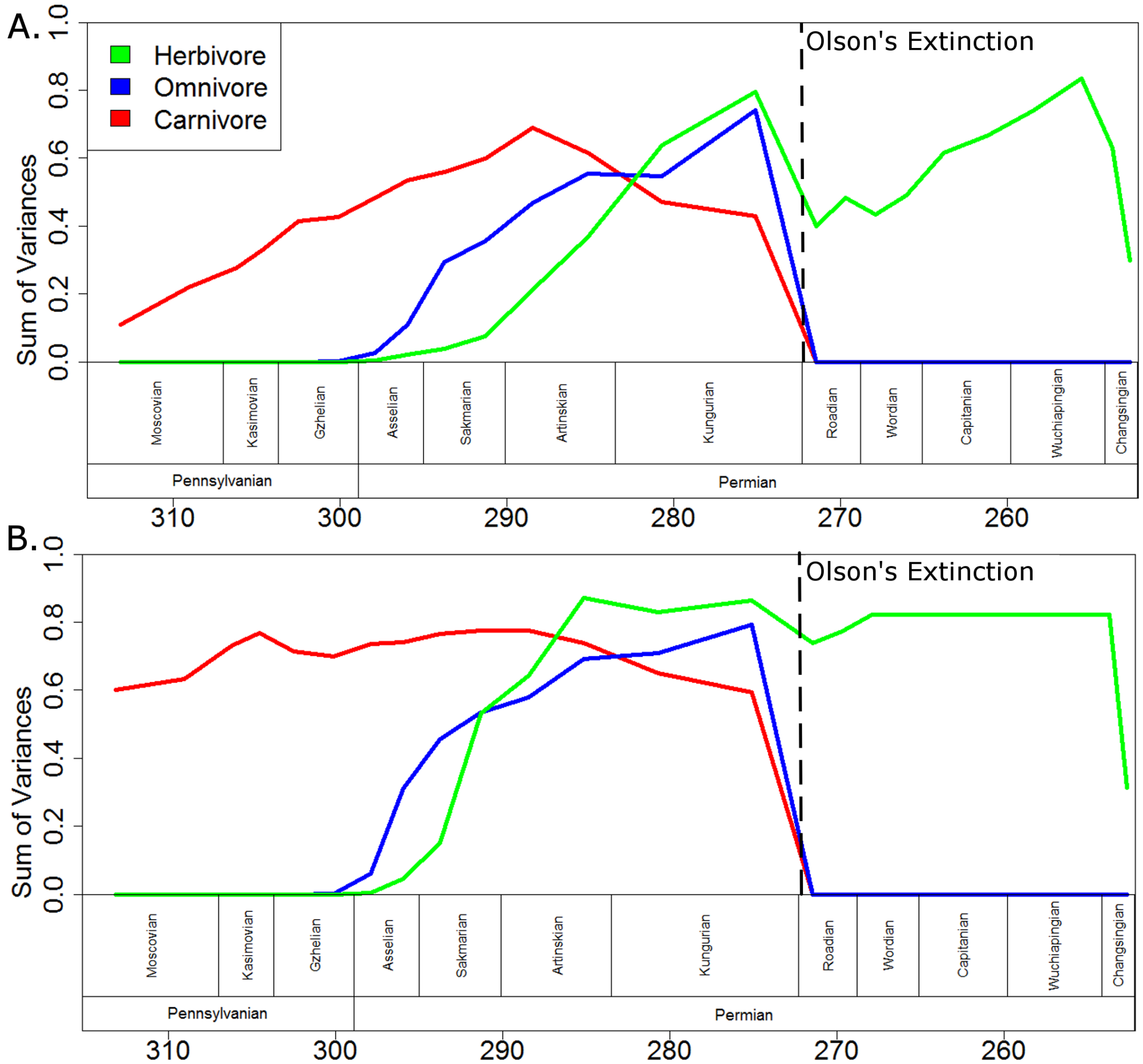

**Figure 7 A comparison of disparity through time of the captorhinids in each dietary regime.** The disparity (sum of variances) calculated for all taxa within each dietary regime in each time bin. Values shown in the graph are the means of the values calculated in 10,000 stochastic maps of dietary evolution. The dashed line represents the mass extinction event dubbed Olson's Extinction. (A) Morphology along each branch calculated assuming a gradualist model of evolution; (B) Morphology along each branch calculated assuming a punctuated model of evolution.

along herbivorous branches that significant increases in rates of morphological evolution are identified in the majority of the 100 time calibrated trees, and in the overwhelming majority of stochastic maps the mean rate of evolution in herbivorous lineages is higher not only than in the other dietary categories but crucially is also higher than in randomly selected clusters of taxa with an equal sample size in more than 94% of the stochastic maps. Further support for higher rates of evolution among herbivorous captorhinids than in other dietary regimes can be found in the lineage leading to *Labidosaurus*; a reversal from a herbivorous ancestor to an omnivorous taxon usually coincides with a significant decrease in rates of evolution. The herbivorous captorhinids also occupy a wider range of morphologies than the other dietary categories, indicating that the increased rate of evolution was an exploration of new morphologies, not simply re-entering established

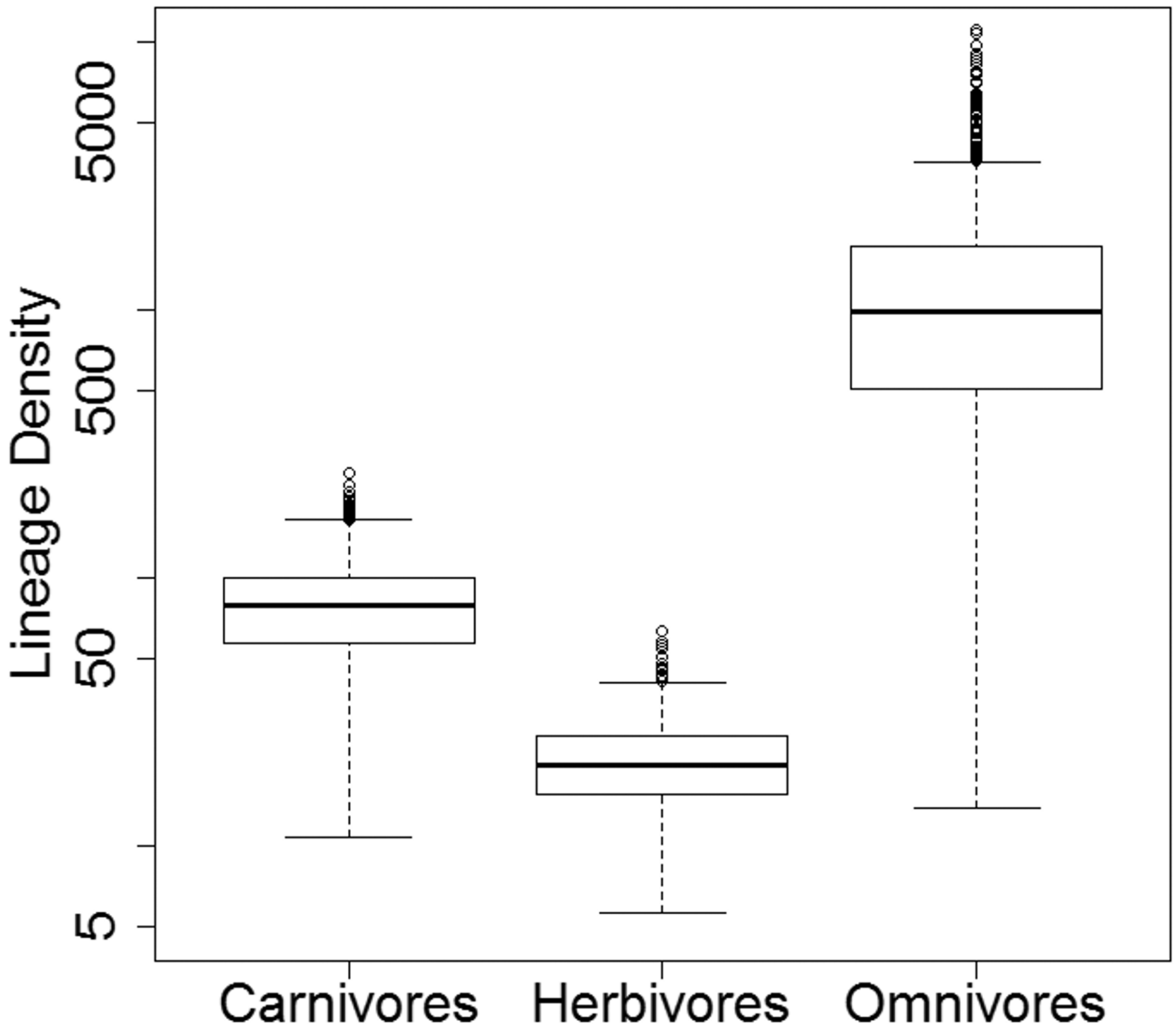

**Figure 8 Lineage densities of captorhinids in each dietary regime.** Boxplots indicating the distribution lineage densities of the captorhinids in each dietary regime, calculated in each of the 10,000 stochastic maps of dietary evolution.

regions of morphospace. This inference is supported by them consistently showing a lower lineage density than either of the other two dietary regimes. When one examines the phylomorphospace (Fig. 10), one may observe that the carnivorous lineages are concentrated in a small region: low values of PC1 and PC2. The Omnivores are found at higher values of PC2. The herbivores, however, explore a great range of values along PC1, and represent the extreme values of PC2.

While carnivorous and omnivorous captorhinids both show a gradual increase in disparity up to a peak in the Artinskian and Kungurian respectively, the herbivorous captorhinids show a much more rapid increase in morphological diversity. Herbivorous taxa do not appear in the fossil record until the Kungurian (although calibrating the phylogeny using the Hedman approach indicates an earlier origin), yet by the late Kungurian they already show a greater morphological diversity than either the carnivores or omnivores show at any point in their evolutionary history. Although the disparity of herbivores falls across the Kungurian/Roadian boundary, a trough possibly related to the mass extinction event known as Olson's Extinction (*Sahney & Benton, 2008*; *Brocklehurst et al., 2015*), the

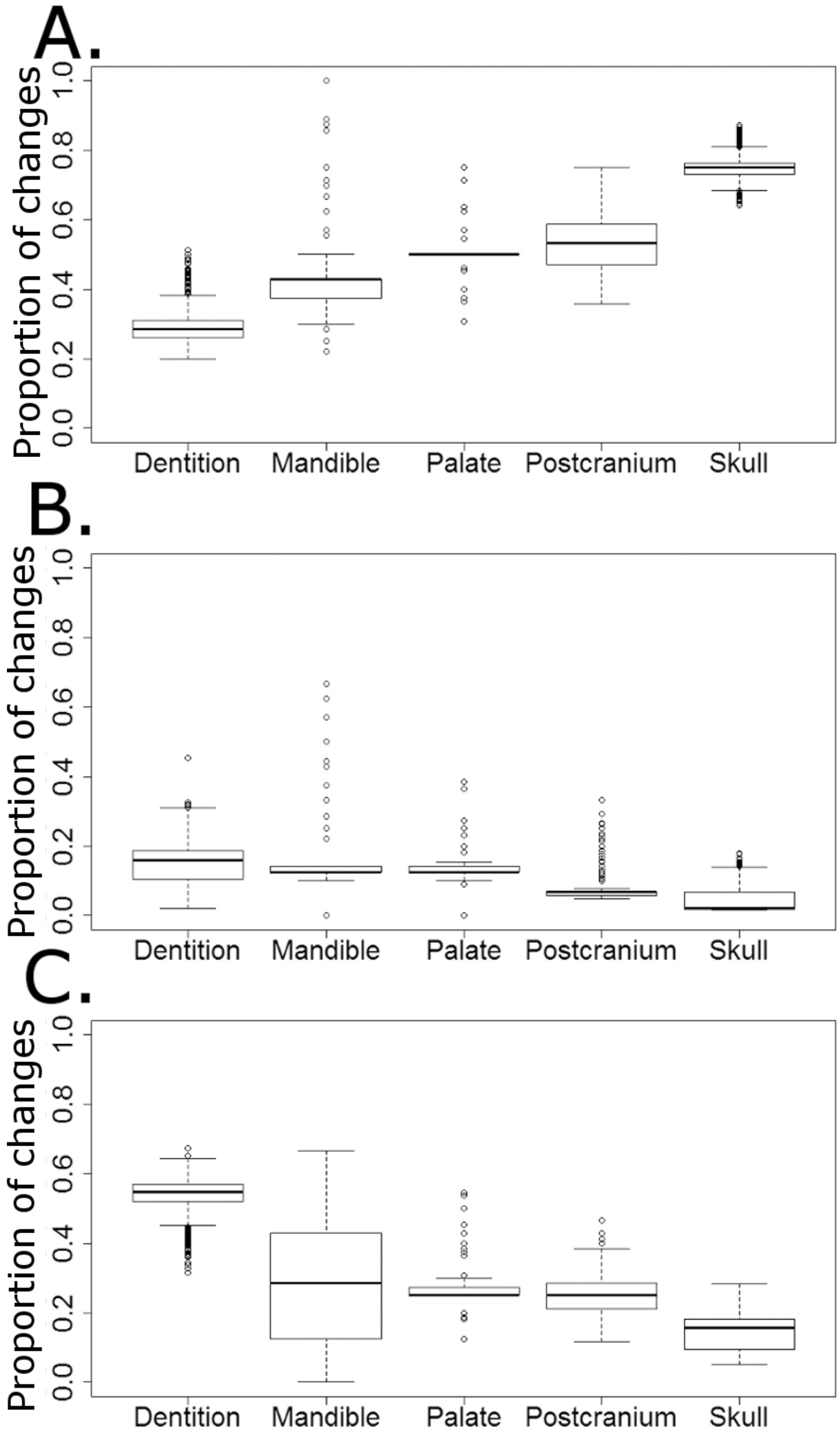

**Figure 9 The proportion of characters within each skeletal region changing within each dietary regime.** Boxplots illustrating the distribution of the proportions of character changes in each skeletal region occurring in each dieatary regime, calculated in each of the 10,000 stochastic maps of dietary evolution. (A) Carnivores; (B) Omnivores; (C) Herbivores.

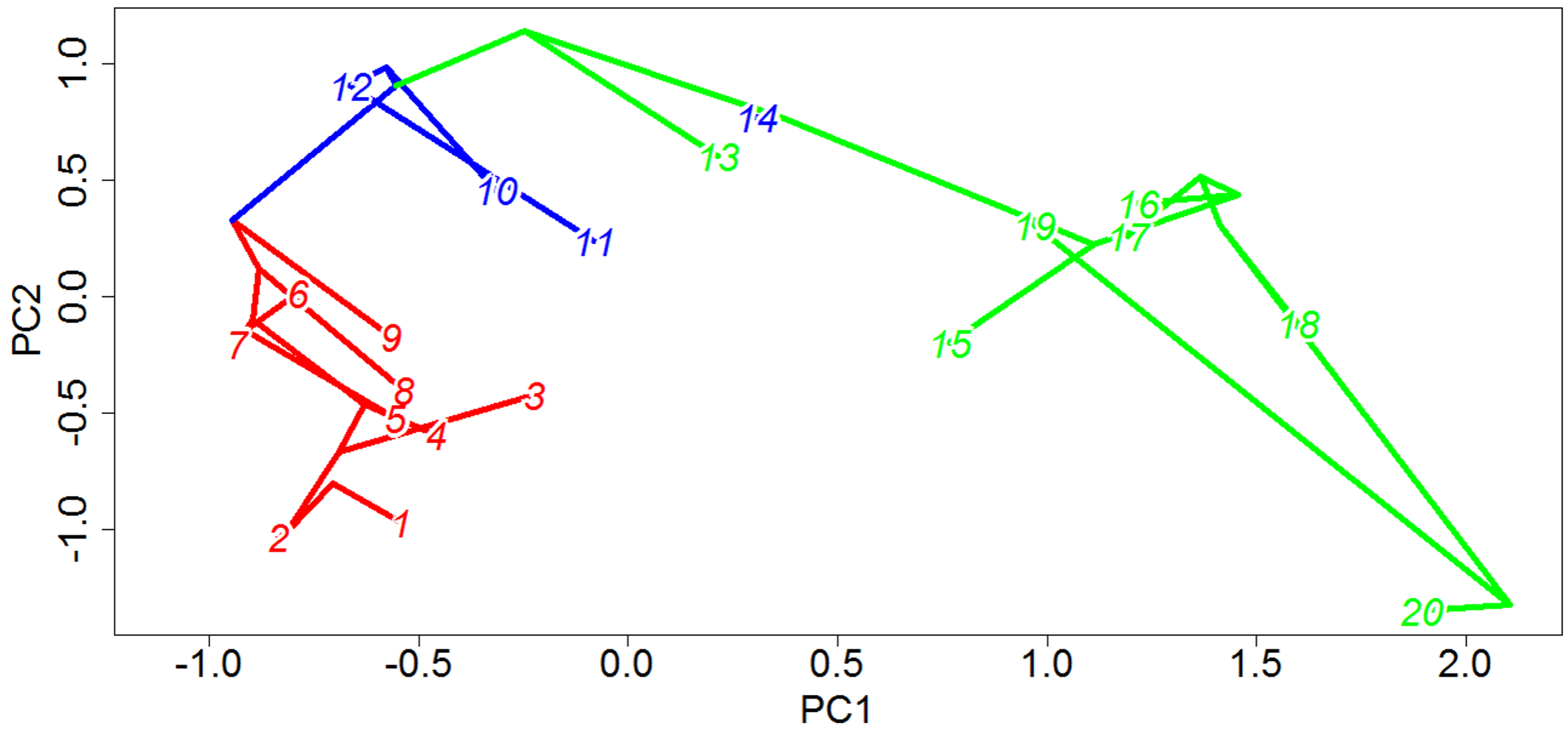

**Figure 10 Phylomorphospace of captorhinids.** The phylogeny of captorhinids plotted over principal coordinates 1 and 2. Colours of lineages represent the die found to have the highest probability by the likelihood ancestral state reconstruction. Taxon labels—1, *Thuringothyris*; 2, *Concordia*; 3, *Opisthodontosaurus*; 4, *Rhiodenticulatus* ; 5, *Reiszorhinus*; 6, *Romeria prima*; 7, *Romeria texana*; 8, *Protocaptorhinus* ; 9, *Saurorictus*; 10, *Captorhinus laticeps* ; 11, *Captorhinus aguti*; 12, *Captorhinus magnus* ; 13, *Captorhinikos chozensis*; 14, *Labidosaurus*; 15, *Captorhinikos valensis* ; 16, *Labidosaurikos*; 17, *Moradisaurus* ; 18, *Gansurhinus*; 19, *Rothianiscus* ; 20, MBCN 15730.

morphological diversity recovers during the Guadalupian and Lopingian, reaching an even higher peak of disparity by the Wuchiapingian. Even though the species richness of captorhinids is substantially decreased by Olson's extinction, the herbivorous lineages continue to show increased morphological innovation. It is possible that the herbivorous captorhinids were more resilient to extinction than others: they make up the majority of the post-extinction diversity, and Middle/Late Permian carnivorous species exhibit low abundances (*Jalil & Dutuit, 1996*; *Golubev, 2000*; *Modesto & Smith, 2001*). Moreover, the late Permian carnivore included in this analysis, *Saurorictus*, appears to experience a significant decrease in rates of evolution in contrast with its herbivorous relatives (Fig. 4). This pattern has been previously observed in Paleozoic and Mesozoic amniotes: the higher species richness of specialist herbivores appears to be driven not by increased origination rates but by higher extinction rates in their close relatives (*Brocklehurst et al., 2015*).

One interesting point to note is that, although herbivores overall show increased rates, disparity and exploration of novel morphotypes than carnivores or omnivores, the significant rate increases appear to be concentrated along the backbone of the herbivorous lineage; in very few of the time calibrated trees are rate increases observed along terminal herbivorous lineages (Fig. 4). This may also be seen in plots of the phylomorphospace (Fig. 10); the greatest changes in morphology occurred along the lineage leading to Moradisaurinae and the lineage leading to the clade containing *Rothianiscus* and the specimen MBCN 15739 (the captorhinid from Mallorca). The only herbivorous terminal branch along which a similar quantity of morphological change is observed is *Rothianiscus*, and this taxon is returning to an already-explored region of morphospace. From this, one

may perhaps infer a slow-down of the diversification rate following the initial rate increase during the first appearance of herbivory, i.e., an early-burst model.

While the coincidence of the rate and disparity increase with the "key innovation" does not necessarily indicate cause and effect, the nature of the morphological changes provides much stronger evidence. It is not only that the rate of character changes increases coinciding with the shift in diet, but it is that the character changes within the herbivores are those referring to the mandible and dentition; that is, the characters related to the feeding apparatus. In the carnivorous captorhinids, the majority of the character changes occur in the skull and the postcranium, while the dentition remains extremely conservative. It is this observation that moves the inference of an adaptive radiation driven by a key innovation beyond one based on the circumstantial evidence discussed above. The evolution of a herbivorous diet occurs alongside not only an increase in the rate of character changes, but a shift in the pattern of the changes. The changes occurring during the adaptive radiation are directly related to the innovation supposedly driving it, a stronger indicator of a causal relationship.

Prior to the evolution of herbivory in captorhinids the overwhelming majority of vertebrate herbivores were large (*Reisz & Fröbisch, 2014*; *Reisz & Sues, 2000*). Edaphosaurids were the most diverse and abundant high-fibre herbivores throughout much of the Pennsylvanian and the Early Permian (*Pearson et al., 2013*; *Reisz & Sues, 2000*), although they go into decline before the end of the Cisuralian. In the latest Cisuralian Hennessey Formation of Oklahoma they are represented solely by some neural spine fragments (*Daly, 1973*), whilst the only supposed edaphosaurid from the contemporary Clear Fork Group of Texas was recently re-described as an indeterminate moradisaurine captorhinid (*Modesto et al., 2016*).

It has been suggested that edaphosaurids and mordaisaurine captorhinids were occupying similar ecological niches (*Modesto, Lamb & Reisz, 2014*); they both convergently evolved similar strategies to deal with plant material (upper and lower tooth-plates and a propalineal motion of the lower jaw). The possibility of competition has been mooted (*Modesto, Lamb & Reisz, 2014*), with the moradisaurines replacing the edaphosaurids. However, *Modesto et al. (2016)* rejected this due to the limited stratigraphic overlap between the two. Moreover, while edaphosaurids show selection towards larger body size (*Reisz & Fröbisch, 2014*; *Brocklehurst & Brink, 2017*), the herbivorous captorhinids show a greater tendency towards decreases in body size than increases (*Brocklehurst, 2016*), possibly indicating niche partitioning instead of competition. During the latest Cisuralian genera such as *Captorhinikos* and *Labidosaurikos* become the most abundant small herbivores (*Brocklehurst et al., in press*), rather than replacing edaphosaurids as large herbivores.

Instead of viewing them as supplanting edaphosaurids, *Modesto et al. (2016)* suggested that the changing climate of the time was responsible for the radiation of the moradisaurine captorhinids. It is true that the radiation of the Moradisaurinae does coincide with a shift towards a warmer, drier, more seasonal climate, and the captorhinids continue to thrive in the arid equatorial regions for the rest of the Permian (*Dutuit, 1976*; *Ricqlès & Taquet, 1982*; *O'Keefe et al., 2005*; *Brocklehurst et al., in press*), in contrast to their rarity in temporal regions. However, the analysis of rates through time casts doubt on this explanation.

An extrinsic driver of increased morphological diversity, such as climate change, should produce a rate shift during a specific interval of time rather than in a specific clade. The data presented here, on the other hand, suggests no significant increase in rate during the Kungurian. In fact, in all the time calibrated phylogenies a constant rate through time best fits the observed data. The shifts in rate occur along specific branches, not during a specific interval, and therefore must be associated with an intrinsic cause.

It is therefore considered more likely that the shift in diet is the cause for the adaptive radiation; specifically, the shift into the "small herbivore" niche that did not require competition with edaphosaurids, caseids and diadectids. Although bolosaurid parareptiles did occupy this niche in some areas during the early and middle Permian, they are comparatively rare and exhibit low species richness (*Reisz & Fröbisch, 2014*). The radiation observed in captorhinids represents an expansion into an extremely under-filled region of ecospace, which they could occupy more efficiently than bolosaurids. It is possible that the increased dental and mandibular innovation allowed the captorhinids their greater success. Herbivorous captorhinids possess multiple tooth rows (in some taxa as many as eleven) and the ability to move the jaw propalineally (*Heaton, 1979*; *Dodick & Modesto, 1995*; *Modesto et al., 2007*; *Modesto, Lamb & Reisz, 2014*), creating an effective surface for grinding and shredding plant matter. Other dental and mandibular innovations appearing within the Moradisaurinae include a saddle-shaped occlusal surface of the teeth and a more robust ramus of the jaw.

## CONCLUSIONS

- A single transition to herbivory in Captorhinidae is most found to be most probable, although whether from a carnivorous or omnivorous ancestor is unclear. *Labidosaurus* appears to represent a reversal to an omnivorous diet from a herbivorous ancestor.
- Significant increases in rates of discrete character change are observed coinciding with the origin of herbivory. The herbivorous lineages are found to have higher rates of evolution than their carnivorous and omnivorous relatives.
- The herbivorous captorhinids were more morphologically diverse than their carnivorous and omnivorous relatives, and reached their peak disparity more rapidly.
- The shift to higher rates of discrete character change is accompanied by a shift towards increased evolution of the mandible and dentition, supporting a causal link between the origin of a herbivorous diet and the radiation observed in captorhinids during the Kungurian.

## ACKNOWLEDGEMENTS

I would like to thank Graeme Lloyd for helpful discussion and assistance with the R package Claddis. Roger Close offered comments on an early draft of the manuscript. Martin Ezcurra, Manabu Sakamoto and an anonymous reviewer made many helpful suggestions which greatly improved the manuscript.

### Funding

This study was funded by a Deutsche Forschungsgemeinschaft grant (number FR 2457/5-1) awarded to Professor Jörg Fröbisch. The funders had no role in study design, data collection and analysis, decision to publish, or preparation of the manuscript.

### Grant Disclosures

The following grant information was disclosed by the author:
Deutsche Forschungsgemeinschaft grant: FR 2457/5-1.

### Competing Interests

The authors declare there are no competing interests.

### Author Contributions

- Neil Brocklehurst conceived and designed the experiments, performed the experiments, analyzed the data, wrote the paper, prepared figures and/or tables, reviewed drafts of the paper.

### Data Availability

The raw data has been supplied as Supplementary Files.

### Supplemental Information

Supplemental information for this article can be found online at http://dx.doi.org/10.7717/peerj.3200#supplemental-information.

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
