# Peer review of "Rates of morphological evolution in Captorhinidae: an adaptive radiation of Permian herbivores"

_PeerJ, doi:10.7717/peerj.3200_

## Round 0.1 · original submission · Major Revisions

First of all, i wanted to apologize for my delayed decision, but i was waiting for the last review and wanted to comprehensively go through your manuscript once more taking into account all reviews. In summary, I see great potential and merit in the analysis of changes in morphological disparity and evolutionary rates during the evolutionary history of the captorhinids and their relationship with dietary shifts In addition, i greatly appreciate the use of quantitative methods to esimate ancestral morphologies and morphological diversity. However, there are still some crucial points which need to be implemented/addressed before publication. As some of these involve additional analyses, i will need some time and effort in line with my major revisions decision The main points which need to be addressed are:

Definition of an adaptive radiation: You use a quite traditional definition of an adaptive radiation – which was introduced at a time when all radiations were considered to be adaptive: compare Erwin (1992); Simões et al. (2016). Morphological diversity and disparity is not necessarily tightly linked; furthermore, the relative role of key innovations and ecological opportunities in driving such diversifications is still debated (Losos and Mahler, 2010; Yoder et al., 2010; Lieberman, 2012; Erwin, 2015). Please discuss this issues in more detail and define in more detail what you consider to be an adaptive radiation in this context (see also comments by reviewers 1 and 2) and how it could be most tested using published literature (see also comments by reviewers 1 and 2).

Small sample size: your sample size is quite low (20 tips in phylogeny) – which is not uncommon when focusing on certain fossil groups. Nevertheless, the manuscript would benefit from discussing the potential pitfalls for your results and interpretations in more detail (how are rates effected by sample size). This could be partially mitigated by doing sensitivity analyses (see comments by reviewer 1), but this might be hard to do in case of really low sample size. I strongly urge the author to validate the obtained results by using alternative (Bayesian) approaches for your particular dataset and questions (see comments by reviewer 2).

Bayesian Methods: I believe the manuscript would significantly benefit from implementing Bayesian approaches which are fast for small datasets and more appropriate as you do not have to assume different rate models which could bias small dataset (see particularly the comments by reviewer 2). I think it still makes sense to keep in the results obtained with your current approach and compare both approaches for validation.

Material and Methods section: This section needs more detail and discussion, particularly concerning the novelty of the approach (see comments by reviewer 1 and 3). Literature references with respect to the used and relevant methodology such as Reversible Jump MCMC approaches or “stochastic mapping” needs to be added (see also comments by reviewer 2). I strongly advice the author to also make the used R-scripts available so everybody can reproduce your results and apply them to their own datasets.

Citation of manuscripts in review: please remove these as this cannot be verified yet

In addition to suggestions by the reviewers, please also address the following:
Line 59: How can they be sure that most were detritivorous ? This is kind of like the null hypothesis, but it might have looked quite different if more data become available. Could you cite some more references in this context? An alternative would be to write “were probably detritivorous” or “have been estimated to be detritivorous”
Line 64: Please provide some reference for the presence of “arthropod herbivores”
Line 68-71: The Simpson´s model is a bit superseded in the meantime – not all morphological or taxonomic diversification are necessarily adaptive radiations (Erwin, 1992; Simões et al., 2016); please all discuss the latest development of the definition of adaptive radiations and their controlling factors (Losos and Mahler, 2010; Yoder et al., 2010; Lieberman, 2012; Erwin, 2015) as this might be relevant for your purposes. Not only evolutionary novelty (“key innovations”), but also antagonist extinction or ecological opportunity might be important factors.
Line 81: again, there are several alternative models to Simpsonian adaptive radiations
Line 97: Do you mean Lloyd et al. 2012 or rather 2016?
Line 123: I guess you mean the oldest “reliably” dated maximum age?
Line 261: it might be better to write “all 9986 of the cases”
Line 294-296: Isn´t this rather circular, if herbivory or feeding in general is mainly based on dental characters and mandibular characters?
Line 328: you cannot cite articles which are not yet published or at least in press. Please remove “Brocklehurst et al. in review”
Line 355-356: Please provide reference for competition hypothesis (moradisaurines replacing edaphosaurids).
Line 370-371: do you have a reference to corroborate this “produce a shift at a specific point rather than in a specific clade”
Line 378: Could they are also be certain characters which could have made bolosaurids less effective?
Figure 3 and 4: It would be useful to have a comparison with bolosaurid diversity for comparative purposes
Figure: Please add the timing of “Olson´s gap” and/or Permian-Triassic extinction to graph (e.g., with a stippled line)

Suggested references:
Erwin, D. H., 1992, A preliminary classification of evolutionary radiations: Historical Biology, v. 6, no. 2, p. 133-147.
Erwin, Douglas H., 2015, Novelty and Innovation in the History of Life: Current Biology, v. 25, no. 19, p. R930-R940.
Lieberman, B. S., 2012, Adaptive Radiations in the Context of Macroevolutionary Theory: A Paleontological Perspective: Evolutionary Biology, v. 39, no. 2, p. 181-191.
Losos, J. B., and Mahler, D. L., 2010, Adaptive radiation: the interaction of ecological opportunity, adaptation, and speciation, in Bell, M. A., Futuyma, D. J., Eanes, W. F., and Levinton, J. S., eds., Evolution since Darwin: the first 150 years, Sinauer, p. 381-420.
Simões, M., Breitkreuz, L., Alvarado, M., Baca, S., Cooper, J. C., Heins, L., Herzog, K., and Lieberman, B. S., 2016, The Evolving Theory of Evolutionary Radiations: Trends in Ecology & Evolution, v. 31, no. 1, p. 27-34.
Yoder, J. B., Clancey, E., Des Roches, S., Eastman, J. M., Gentry, L., Godsoe, W., Hagey, T. J., Jochimsen, D., Oswald, B. P., Robertson, J., Sarver, B. A. J., Schenk, J. J., Spear, S. F., and Harmon, L. J., 2010, Ecological opportunity and the origin of adaptive radiations: Journal of Evolutionary Biology, v. 23, no. 8, p. 1581-1596.

Reviewer 1 ·

Basic reporting

No comment

Experimental design

No Comment

Validity of the findings

No Comment

Additional comments

The author presents an interesting case study of an ‘adaptive radiation’ being linked to the acquisition of a ‘key characteristic’ (herbivory) in the fossil record. The results indicate a link between this acquisition and a move to higher disparity. I feel the manuscript is well-presented and of interest, and should be published. However, I do have some concerns I would like to see addressed.

My main concern is the ability to infer results from the small sample size (n ~ 20 tips in the phylogeny). I think it is important to include some discussions and sensitivity tests to elucidate how well these methods perform with this size of dataset. For example, are they prone to higher error rates with small sample size? Additionally, there are many definitions of adaptive radiations (please see below) but the author favours an interpretation of high rates of morphological evolution being found in all lineages subsequent to the adaptation. Again I feel this conclusion would be firmer with some validation/sensitivity. My main points are summarised below:

1. Dataset Size/sensitivity tests. I think it would be worth validating the results from the Claddis rates analyses by running sensitivity tests, especially as this is a small dataset. An approach similar to Close et al. (2015) could be used. For example, to validate the results the author could use jackknifing of the sample dataset and, although it may be difficult with the small sample size, exclude highly complete taxa in another validation test. These approaches could be employed to show the peak in rates is a real signal (which I believe it probably is).

2. Methods. Some of the Materials and Methods section needs more detail and discussion. Please these my detailed comments for more information

3. Rates. Both the results from Claddis and Stochastic Mapping indicate a high rate of evolution in herbivorous lineages. This is used as evidence for herbivorous lineages having higher rates of evolution. However, how much of this signal of higher rates is driven by the branches leading to the herbivorous lineages (the ‘backbone’ branches on the phylogeny)? Figure 4 seems to indicate these increased rates occur on these ‘backbones’. If these early herbivorous branches are excluded is there an equal distribution of rates between the herbivorous and other lineages? By differentiating the exact points of these high rates it would be possible to see if this adaptive radiation marks move into a novel ecospace (herbivory) or whether this is a sustained radiation within the herbivorous lineages.

This not a major flaw as the there is still a higher disparity in herbivorous lineages (Figure 6) but if these rates are driven by these early branches it could lead to a slight difference in the conclusions. If these high rates are confined to early lineages this dataset may correspond to an “early burst” (Blomberg et al. 2003; Harmon et al. 2010) model in which rates are initially high in the herbivorous clade but then fall through time (here it appears they go to a ‘background rate’).

4. Adaptive Radiations. One issue is what ‘adaptive radiation’ actually means. There has been lots of literature on the subject (e.g., Schluter 2000; Losos 2010), and some authors have argued ‘adaptive radiations’ only refer to increased rates of morphological evolution (Givnish 2015).

The author’s interpretation in the manuscript is acceptable (i.e., increase morphological and diversification rates), but it would be nice to see some discussion of the controversy surrounding what ‘adaptive radiation’ actually means. Furthermore, with the tree containing only a few species, and in which the ‘adaptive’ species are out-numbered, it may be better to class this as an adaptive radiation of morphology alone (Givnish 2015).

Related to that note: is there any link between herbivory and survival? As the carnivorous and omnivorous species seem to go extinct and the herbivorous species endure – has there been discussion about this dietary shift being related to survival?


Specific Comments

Introduction

Line 65-69. Persnickety point. But how would you defined a “somewhat under-filled region of ecospace”

Line 70-71. Here would be a good point to have some introduction on the various definitions of adaptive radiations

Lines 83-89. Is it always the case that acquisition of key traits leads to higher rates of change in the disparity of subsequent lineages? I can see this is the case from some traits (e.g., flowers), but, for example, is this true for continuously growing incisors in rodents?

Materials and Methods

Lines 115. I think use of this method (Hedman 2010; Lloyd et al. 2016) is fine, but was there a reason not to use any of the alternatives such as Cal3 (Bapst 2013)? Would it even be possible to use morphological clock (Ronquist et al. 2012; Lee et al. 2014) to date these trees?

Line 117. Missing word “date”?

Lines 127. Was there a reason to use only 100 replicates rather than 1000?

Line 135-146. Which method was used in the discrete likelihood calculation? Marginal likelihood? Conditional likelihood?

Line 167. Which method/approach was used to apply the stochastic mapping?

Line 172. Typo “phylogeny” should be “phylogenies”

Line 179. Explain the acronym ‘MORD’. How much data is missing? Also, Lloyd (2016) states that this is a good measure to use for missing data but is equivalent to Gower’s Coefficient if data are unordered - are the data here ordered or unordered?

Lines 190-207. I like this approach to incorporating data into the analyses from known phylogenetic analyses. Could you please clarify if this method is used for all branches or terminal branches only? I think it is used here for terminal branches. Would it not be better to use all branches for these calculations? I am not suggestion this method is invalid but I think the manuscript needs more discussion about the chosen approaches, etc.

Results

Lines 224-241. Is this uncertainty in ancestral states incorporated into the rates analyses? For example, the point of which herbivory evolves seems to be contentious. Could the different potential ancestral states be used in the analyses to see how this influences rates?

Line 253-256. As there is a temporal gap between the herbivorous and other taxa, is the unity of temporal rates suggestive of there being no difference in rate between the lineages?

Line 259-261. Missing words after the numbers: “maps”?

Discussion

Lines 308-313. Please see my earlier point – how much of this result is driven by those branches leading to herbivory rather than those within herbivorous lineages?

Lines 332-343. Would we expect these kinds of changes every time a clade moves from to herbivory, or to carnivory from herbivory? It seems this will always reflect a large change in morphology and so possibly high rates

Figures

Figure 2. Could this be given a label on the y-label? Also, is there space to expand the acronyms on the x-axis?

Figure 8. Could this be given a label on the y-label?

References

Bapst, D. W. (2013). A stochastic rate‐calibrated method for time‐scaling phylogenies of fossil taxa. Methods in Ecology and Evolution, 4(8), 724-733.

Blomberg, S. P., Garland, T., & Ives, A. R. (2003). Testing for phylogenetic signal in comparative data: behavioral traits are more labile. Evolution, 57(4), 717-745.

Close, R. A., Friedman, M., Lloyd, G. T., & Benson, R. B. (2015). Evidence for a mid-Jurassic adaptive radiation in mammals. Current Biology, 25(16), 2137-2142.

Givnish, T. J. (2015). Adaptive radiation versus ‘radiation’ and ‘explosive diversification’: why conceptual distinctions are fundamental to understanding evolution. New Phytologist, 207(2), 297-303.

Harmon, L. J., et al. (2010). Early bursts of body size and shape evolution are rare in comparative data. Evolution, 64(8), 2385-2396.

Hedman, M. M. (2010). Constraints on clade ages from fossil outgroups. Paleobiology, 36(01), 16-31.

Lee, Michael SY, Andrea Cau, Darren Naish, and Gareth J. Dyke. Morphological clocks in paleontology, and a mid-Cretaceous origin of crown Aves. Systematic Biology (2014): 63(3), 442-449.

Lloyd, G. T., Bapst, D. W., Friedman, M., & Davis, K. E. (2016). Probabilistic divergence time estimation without branch lengths: dating the origins of dinosaurs, avian flight and crown birds. Biology Letters, 12(11), 20160609.

Lloyd, G. T. (2016). Estimating morphological diversity and tempo with discrete character‐taxon matrices: implementation, challenges, progress, and future directions. Biological Journal of the Linnean Society, 118, 131-151.

Losos, J. B. (2010). Adaptive radiation, ecological opportunity, and evolutionary determinism. The American Naturalist, 175(6), 623-639.

Ronquist, F., Klopfstein, S., Vilhelmsen, L., Schulmeister, S., Murray, D. L., & Rasnitsyn, A. P. (2012). A total-evidence approach to dating with fossils, applied to the early radiation of the Hymenoptera. Systematic Biology, 61(6), 973-999.

Schluter, Dolph. The ecology of adaptive radiation. OUP Oxford, 2000.

·

Basic reporting

The writing can benefit from more clarity and unambiguity. Though overall the English is fine and grammatical errors do not stand out, there are numerous instances of careless mistakes such as repeated words or spelling mistakes, giving me the impression that this manuscript is still an earlier draft, lacking refinement.

Literature references are sufficient for the palaeontological aspects such as the relevant taxonomic literature or past extinction or climatic events, but is lacking with respect to relevant methodology such as Reversible Jump MCMC approaches implemented in programs like BayesTraits. Further, “stochastic mapping” lacks citation, and I am left wondering if this refers to Bollback’s SIMMAP (2006 BMC Bioinformatics 7) including Liam Revell’s implementation in his phytools R package, or if this is the author’s own invention?

Further, the relevant citations with respect to evolutionary interpretation – e.g. adaptive radiation, rates of evolution – are lacking. I think the author should cover and discuss what evolutionary biologists have written on the subjects of adaptive radiation, diversification and rates of trait evolution using modern quantitative approaches. Some representative references for adaptive radiations include the body of work done by Dan Rabosky on diversification slowdowns (which can be interpreted as a form of adaptive radiation), or reviews (Lieberman, 2012. Evol Biol 39: 181-191; Moen and Morlon, 2014. Trends Ecol Evol 29: 190-197; Simoes, 2016. Trends Ecol Evol 31: 27-34). A shameless plugin for my own work but Sakamoto et al. 2016 (PNAS 113: 5036-5040) also showed increases in net speciation associated with key dental and mandibular innovations – i.e. dental battery – as well as a more recent study on hadrosaur dental disparity showing something similar (Strickson et al., 2016. Sci Rep 6: 28904). Some good starting points for modern methods and interpretations of rate heterogeneity include Venditti and Pagel (2010. Trends Ecol Evol 25: 14-20), Venditti et al. 2011 (Nature 479: 393-396) or Baker et al. (2015. PNAS 112: 5093-5098, 2016. Biol J Linn Soc 118: 95-115).

The article conforms to an acceptable standard format. I think there are more figures than necessary – some can be combined for instance. Histograms (e.g. Figs 5 and 6) can be combined with the phylogeny figs (Figs 3 and 4 respectively) – Baker et al. (2015. Biol J Linn Soc) has some good examples. Overall figures look like software output and can be touched up to look nicer – but perhaps this comes down to taste and would not affect the content of the fig. However, the manuscript can definitely benefit with better-explained figure captions.

The manuscript is self-contained, representing an appropriate unit of publication, and does not rely on any preceding work or parallel submissions.

Experimental design

I think the aims set out in this piece of research are original and significant, with a well-defined research question that is relevant and meaningful – the appearance of herbivory in early tetrapods and how that came about is an important evolutionary question worthy of further scrutiny.

However, I cannot say confidently that the investigation has been conducted to the highest technical standard using the methods most appropriate to answer the main research question. I will outline this below:

[Phylogeny]
Even though the author has a phylogenetic dataset and date ranges for the tips, the method chosen is to post-hoc scale branch lengths of undated parsimonious tree(s) – I am aware that this is a fairly common practice in palaeontological studies (I myself use similar methods for very large composite trees where datasets are not available), but given that Bayesian methods are readily available for simultaneous topology inference and divergence/tip dating, and that the author already possesses the necessary data, i.e. data matrix and date ranges, there shouldn’t be any excuse not to infer and date the phylogeny. This way, one will result with a statistically robust posterior sample of dated trees, which is more meaningful than a random collection of stochastically scaled trees. This is because the dates on Bayesian trees are guided by likelihood given a model of evolution and the data, while stochastically scaled trees are produced from randomly sampling dates from tip ranges, the combination of dates of which being astronomical in number. Thus selecting 100 or 1000 random combinations of dates does not cover anywhere near the full extent of possible tree-space. One might as well select one tree scaled to the maximum branch lengths possible and another scaled to the minimum branch lengths possible, and reporting on the two extreme points.

A posterior sample of Bayesian inferred trees is statistically robust and models can either be integrated over that sample of trees or one can choose the maximum clade credibility tree to fit a single representative model of evolution.

[Dietary evolution]
I am surprised a Bayesian approach is not taken here. Given the small sample size, a Bayesian model can be fit in minutes. The benefit of a Bayesian approach is that one needs not fit three different models of rate evolution (equal rates, symmetric, asymmetric) and compare using AICc or the likes, but RJ MCMC can actually allow for different combinations of character states to have different transition rates. Further using programs like BayesTraits, one can simultaneously infer ancestral states at nodes of interest and estimate transition rates.

Further, judging from Fig 3 and how the dietary categories are distributed at the tips, there are only a few nodes that should interest us with respect to ancestral states and associated probabilities. I think the author can just run BayesTraits and reconstruct ancestral states for just those few nodes. The documentation for BayesTraits is straightforward so it shouldn’t be a huge endeavor.

[Analysis of rate variation]
If one were to infer the tree and date it simultaneously using software like BEAST, branch-wise rates of character change will be part of the output. Such rates are guided by likelihood given a model of evolution and the data, and are statistically robust.

In my opinion, summarising branch-wise rates into binned time series is problematic, as the time series will likely be biased by phylogenetic non-independence. However, I concede that this is a contentious point.

Perhaps a minor point but the paragraph describing stochastic mapping is vague, without citations and confusing.

[Disparity]
How are the ancestral states reconstructed prior to calculating morphological distances? I think this will heavily bias the ordinated coordinates. Regardless, I am rather skeptical of including ancestral reconstructions in morphospace constructions. There are some uninterpretable artifacts associated with this approach – you can clearly see this in Brusatte et al.’s (2011) morphospace plot (but also in Sakamoto, 2010. Proc R Soc B 277: 3327-3333) in which ancestral nodes plot outside of the range of the regions of morphospace occupied by the descendants. I don’t think this should happen given that ancestors are inferred from the tips so are necessarily some weighted average of the descendants – i.e. nodes should fall within descendant range – but it does, so there is clearly something wrong with this approach. I think more testing needs to be done. Perhaps variable rates need to be incorporated into the ancestral reconstruction (using a stretched tree would suffice)?

Why are you using sum of variances? If I remember correctly, you’d want to look at sum of ranges or root product of variances – sum of variances is a strange one that I’ve always had difficulty interpreting.

I also think a morphospace (or phylomorphospace) plot needs to be presented in the relevant section, especially when discussing how herbivorous taxa are expanding into new regions of morphospace instead of utilizing the same region of morphospace (L318-319) – actually, Sidlauskas (2008. Evolution 62: 3135-3156) introduced a metric, which he called Lineage Density that measures how clades are occupying phylomorphospace, that may be of interest.

Overall, I don’t think the methods are explained well. In particular, what the author calls stochastic mapping is not sufficiently explained and lacks citations.

Validity of the findings

For the reasons I’ve stated above I don’t think the results are statistically sound – though others may disagree. I am being strict here, in that given that I don’t think the best methods are used here, the results are therefore not as robust as they could be.

Furthermore, linking the timing of a rate shift and transition to herbivory is circumstantial and not statistically tested. As the author is trying to determine causation, a simple and elegant way to do this would be to model the effect of dietary regime on rates of character evolution using phylogenetic regression (or phylogenetic ANOVA). If herbivory is truly associated with increased rates, then the phylogenetically corrected mean rate would be higher than the other regimes. In this sense I don’t think the conclusions – that rates increase with the origin of herbivory – are supported statistically.

I don’t think results from the disparity analyses are tied in well with the other two analyses, and to me doesn't add much to answer the research question. Expanding clades would be expected to have increasing variance through time, so compared to carnivores and omnivores that wane through time, herbivores continue to maintain high variance in trait data. A series of time sliced phylomorphospaces would convey the same information more visually stunningly…

While “adaptive radiation” is mentioned several times, the author doesn’t exclusively model or test adaptive radiations, e.g. diversification slowdowns. Adaptive radiations are characterised by an initially high rate of diversification followed by a slowdown as niches are filled.

Additional comments

I think this study as it currently stands is overly complicated, confusing and introduces a lot of pseudo-statistical procedures that can be replaced by more elegant, statistically robust methods.

Also, there are too many figures for the length of the manuscript. I think they can be combined or simplified: Is it necessary to show two trees? There doesn't seem to be any rational reason other than arbitrary choice for which two trees to present, so why not just present one? If a Bayesian posterior sample of trees were generated, then a maximum clade credibility tree can be used as a representative tree...

·

Basic reporting

The manuscript of Brocklehurst analyse the macroevolutionary history of an interesting group of Palaeozoic sauropsids, the captorhinids. The analyses are focused on the morphological disparity and evolutionary rates during the evolutionary history of the group, which includes dietary shifts from carnivorous to herbivorous species. The analyses conducted in this manuscript are comprehensive and well conducted, adding important and interesting information to characterize evolutionary patterns in the Captorhinidae. In addition, the author implement a new method to estimate ancestral morphologies using a gradual model of evolution in order to correct morphological disparity estimations from incompleteness in the fossil record, which in my opinion it is a step forward in this kind of analyses.
I did not notice major problems in the manuscript, which is well-written, illustrated, and the references are comprehensive. I made only minor comments and suggestions in an edited version of the manuscript (attached here). In addition, I detailed below four points that represent the major observations that I have about the manuscript. I strongly suggest the author to check and improve the manuscript based on these four points (specially points 2 and 4):
- 1. Lines 328, 362, and 368: it is generally not accepted to quote manuscripts under review. The editor should consider if the cite of Brocklehurst et al. (in review) in this part of the manuscript is allowed.
- 2. Discussion and Conclusions: one of the main novelties of the manuscript is the method to estimate morphologies unsampled in the fossil record, but indicated to be present during the time bin by the phylogenetic topology, using a gradual model of evolution. I strongly suggest the author to discuss deeper the benefits of this new methodology over previous proposals (e.g. Brusatte et al. 2010 and the function implemented in the package claddis for R [Lloyd 2016]), as well as possible caveats (e.g. assume a constant model of evolution if the sampled morphology of the group suggested a non-constant, e.g. early burst, model of evolution).
- 3. Figure 1: this figure does not have a legend explaining what the vertical axes account for. I infer that it is morphology, but the author should clarify it.
- 4. Supplementary Information: I strongly suggest the author to include an additional supplementary file with the R code that it was used to conduct the analyses and, specially, the code to estimate ancestral morphologies using a gradual model of evolution. These files will help other researchers to replicate and use the methodologies employed in this manuscript.

In conclusion, I strongly recommend the publication of this manuscript in PeerJ after minor modifications.

Experimental design

The experimental design of the manuscript is well conducted and follows that of previous works. In addition, it is proposed in the manuscript a new methodology that in my opinion will be welcomed by other researchers (see basic reporting).

Validity of the findings

The results of the analyses are novel, robust, and well discussed.

Additional comments

I congratulate the author for the incorporation of a gradual model of evolution at the time of estimate morphological disparity including ancestral morphology estimates. I was planning to work on this method and I am happy that someone already implemented it!

---

## Round 0.2 · Minor Revisions

Thanks you for addressing our suggestions. The manuscript greatly benefitted from implementing additional analyses and explaining the rationale and differences behind these approaches in more detail. Making available your R-scripts is also greatly appreciated and in line with the PeerJ policy. As far as I am concerned the manuscript is as good as accepted. I just found some additional minor (mostly formatting) issues in the reworked version, which I would like you to still address. It would normally be customary to also address the editor recommendations 1-by-1 in the rebuttal which would be something to consider when you would submit again to PeerJ. I am looking forward to the final version of your manuscript.

---

## Round 0.3 · accepted · Accept

Thank you for integrating these final suggestions. It has been a real pleasure to handle your manuscript.